# Controlled Synthesis of Cs_2_NaYF_6_: Tb Nanoparticles for High-Resolution X-Ray Imaging and Molecular Detection

**DOI:** 10.3390/nano15100728

**Published:** 2025-05-12

**Authors:** Jian Zhao, Kunyang Wang, Wenhui Chen, Deyang Li, Lei Lei

**Affiliations:** Institute of Optoelectronic Materials and Devices, China Jiliang University, Hangzhou 310018, China; zhaojian202411@163.com (J.Z.); 19357157434@163.com (W.C.); lideyang@cjlu.edu.cn (D.L.); leilei@cjlu.edu.cn (L.L.)

**Keywords:** fluoride, XEOL, nanoscintillator

## Abstract

Rare-earth-doped fluoride nanoparticles (NPs), known for their tunable luminescence and high chemical stability, hold significant potential for applications in X-ray imaging and radiation dose monitoring. However, most research has primarily focused on lanthanide-doped NaLuF_4_ or NaYF_4_ nanosystems. In this work, Cs_2_NaYF_6_:Tb NPs with enhanced X-ray excited optical luminescence (XEOL) intensity were developed. Our results indicate that low oleic acid (OA) content and a high [Cs^+^]/[Na^+^] ratio favor the formation of pure cubic-phase Cs_2_NaYF_6_:Tb NPs. Cs_2_NaYF_6_:Tb NPs were successfully fabricated into thin films and employed as nanoscintillator screens for X-ray imaging, achieving a high spatial resolution of 20.0 Lp/mm. Beyond X-ray imaging applications, Cs_2_NaYF_6_:Tb NPs were also explored for spermine detection, demonstrating high sensitivity with a detection limit of 0.44 μM (under X-ray excitation) within a concentration range of 0–60 μM. These findings may contribute to the development of novel lanthanide-doped fluoride nanoscintillators for high-performance X-ray imaging and molecular sensing.

## 1. Introduction

Scintillators are capable of converting high-energy X-rays into low-energy visible photons. Scintillator-based X-ray imaging technology is of great interest in various applications like industrial flaw detection [1], medical imaging [2,3], and safety inspection [4]. Most commercial scintillators are mainly bulk materials, such as CaWO_4_ [5], NaI:Tl [6], (Lu,Y)_2_SiO_5_:Ce (LYSO:Ce) [7], and Bi_4_Ge_3_O_12_ [8]. Nevertheless, the synthesis and preparation of these scintillators often need harsh conditions such as high temperature and pressure, which greatly increases the preparation costs [9]. In addition, it is hard to tune the X-ray excited optical luminescence (XEOL) wavelengths in these systems for different practical applications.

Nanoscintillators, such as perovskites and fluorides, have attracted widespread attention in the field of X-ray imaging due to their facile synthesis and tunable XEOL wavelengths. Compared to halide perovskites, lanthanide-doped fluoride nanoparticles exhibit significantly higher stability [10]. Lanthanide activators possess abundant excited energy levels, enabling tunable emissions from ultraviolet to near-infrared under X-ray irradiation [11,12]. Lanthanide activators possess abundant excited energy levels in many ways, such as ion co-doping technology [13,14], and the construction of core-shell (CS) structures [15] has been used in enhancing XEOL performance. For example, through co-doping an appropriate concentration of Gd^3+^ ions into NaLuF_4_:15Tb NPs, the XEOL intensity is approximately five times stronger owing to the energy transfer from Gd^3+^ to Tb^3+^ [16]. Furthermore, coating the NaYF_4_ shell out of NaLuF_4_:Tb NPs can enhance the XEOL intensity by a factor of 1.5, which is attributed to the passivation of surface quenchers [17]. In general, materials with a high atomic number (Z) and density (ρ) have a large X-ray attenuation coefficient (μ). The relationship can be expressed by the following equation:μ = ρZ^4^/AE^3^
where A is the atomic mass, and E is the X-ray photon energy [18]. Therefore, incorporating heavy metal elements into a host is a typical route to improve XEOL intensity [19]. For instance, Ou et al. [17] demonstrated that the XEOL intensity of NaLuF_4_:15Tb (Z = 71) NPs is two times higher than that of NaYF_4_:15Tb (Z = 39) NPs, indicating that using heavy atoms benefits the achievement of strong XEOL luminescence. Therefore, theoretically, incorporating high-Z Cs^+^ ions can increase the XEOL intensity of lanthanide-doped fluoride NPs. In addition, compared with other high-Z host materials (e.g., Ba-based fluoride systems), this structure exhibits lower biotoxicity and higher luminescence efficiency [20]. For example, Yang et al. [21] synthesized a bulk phosphor of Cs_2_NaYF_6_:Pr, which has an ultraviolet emission power density of more than 10 mW/m^2^ and is widely used in ultraviolet sterilization and other fields. However, the XEOL performances of lanthanide-doped Cs_2_NaYF_6_ nanosystems is rarely reported.

In this work, the Cs_2_NaYF_6_: Ln^3+^ nanoscintillators with improved XEOL intensity were synthesized. Our results indicate that low oleic acid (OA) content and a high [Cs^+^]/[Na^+^] ratio helped to achieve pure Cs_2_NaYF_6_: Ln^3+^ NPs. The Cs_2_NaYF_6_ NPs doped with Tb^3+^, Eu^3+^, Dy^3+^, Er^3+^, or Tm^3+^ ions presented strong scintillation performance under X-ray irradiation. The XEOL intensity of the Cs_2_NaYF_6_: Tb NPs was much stronger than that of NaYF_4_: Tb NPs. Cs_2_NaYF_6_:15Tb NPs were fabricated into a thin film and used as a nanoscintillator screen for X-ray imaging. Finally, a high spatial resolution of 20.0 LP/mm was realized. Beyond X-ray imaging applications, Cs_2_NaYF_6_:Tb NPs were also explored for spermine detection, demonstrating high sensitivity with a detection limit of 0.44 μM (under X-ray excitation) and 0.36 μM (ultraviolet excitation) within a concentration range of 0–60 μM.

## 2. Materials and Methods

Cs_2_NaYF_6_:Ln (Ln = Tb^3+^, Eu^3+^, Tm^3+^, Dy^3+^, Er^3+^) NPs were synthesized via a well-known thermal decomposition method. Typically, 0.85 mmol of Yttrium Acetate Y(Ac)_3_, 0.15 mmol of Terbium Acetate (Tb(Ac)_3_), and 1 mmol of Sodium Acetate (NaAc) were mixed with 8 mL of oleic acid (OA), 4 mL of Oleylamine (OLA), and 8 mL of Octadecene (ODE) in a 50 mL three-necked round-bottom flask. The resulting mixture was heated to 120 °C under N_2_ flow with constant stirring for 30 min to remove residual water and oxygen. After cooling down to room temperature (RT), 8 mL of methanol solution containing 10 mmol of CsF was added, and the solution was stirred at 70 °C for 90 min. The resulting solution was heated to 320 °C under N_2_ flow with vigorous stirring for 60 min, and then cooled down to RT. The obtained NPs were precipitated by the addition of 20 mL of ethyl acetate. The product solution was centrifuged at 8000 rpm for 5 min. The supernatant was discarded, and the precipitated NPs were re-dispersed in 2 mL of hexane. By centrifugation at 5000 rpm for 5 min, the supernatant was precipitated with 4 mL ethyl acetate. The solution was centrifuged again at 12,000 rpm for 5 min. The final NPs were obtained by discarding the supernatant. Cs_2_Li_0.5_Na_0.5_YF_6_:15Tb NPs were synthesized using a similar method, in which 0.5 mmol of NaAc and 0.5 mmol of Lithium Acetate (LiAc) were used.

For synthesis of the Cs_2_NaYF_6_:15Tb film, first, 100 mg of poly(methyl methacrylate)(PMMA) and 1 mL of toluene were mixed, which then were stirred to form a transparent solution. A total of 75 mg of Cs_2_NaYF_6_:15Tb NPs was added into the above PMMA toluene solution [22,23], which then was stirred for more than 3 h to generate a uniform paste product without agglomerates. The resultant product was transferred into a designed mold, which was then volatilized at room temperature for 12 h until the film was formed.

X-ray diffraction (XRD) analysis was carried out by a powder diffractometer (Bruker D8 Advance, Bruker Corporation, Billerica, MA, USA). The morphology and size of the products were characterized by a field emission transmission electron microscopy (TEM, FEI Tecnai G2 F20, Thermo Fisher Scientific, Eindhoven, The Netherlands) equipped with an energy dispersive X-ray spectroscopy (EDS, Aztec X-Max 80T, Oxford Instruments, Abingdon, Oxfordshire, UK) and a field emission scanning electron microscopy (SEM, Hitachi SU8010, Hitachi High-Tech Corporation, Tokyo, Japan). For the TEM measurements, the NPs were deposited onto a copper grid by drop-casting. Scintillation spectra were recorded on an OmniFluo-Xray-JL system (PMT-CR131-TE detector, 185–900 nm, Zhuoli Instrument Co., Ltd., Beijing, China) with a mini MAGPRO X-ray excitation source. An X-ray imaging system was employed, consisting of an X-ray tube (TUB00154-9A-W06, Hamamatsu Photonics, Hamamatsu, Japan), a lens, an external reflecting prism, and an ORCA-Fusion BT camera (C15440-20UP, Hamamatsu Photonics, Hamamatsu, Japan). The imaging object was placed on the previously prepared scintillating film and irradiated with X-rays at a voltage of 50 kV. The resulting images were captured using the camera.

Inductively coupled plasma optical emission spectrometry (ICP-OES) was performed using an Agilent 720ES instrument, equipped with a high-frequency generator operating at 27.12 MHz to sustain stable plasma. Calibration was carried out using a series of terbium standard solutions, prepared by appropriate dilution in 2% *v*/*v* nitric acid to generate a concentration-dependent calibration curve. All standards were prepared in volumetric flasks and mixed thoroughly to ensure homogeneity. Sample preparation procedures: a 0.5 g sample was added into 10 mL of nitric acid. The resulting solution was transferred to a 20 mL volumetric flask. The digestion vessel was rinsed multiple times with deionized water, and the rinse solutions were added to the flask.

## 3. Results

### 3.1. Characterization of Cs_2_NaYF_6_:15Tb (Mol%) NPs

The method used for synthesis is well-known as the thermal decomposition method. The SEM image depicted in Figure 1a reveals a uniform distribution of the prepared Cs_2_NaYF_6_:15Tb NPs, with irregular morphology. Size distribution histogram analysis reveals that the mean particle size is approximately 55 nm (Appendix A). Figure 1b shows the XRD plot of Cs_2_NaYF_6_:15Tb NPs, and the diffraction peaks of the samples correspond to standard data (JCPDS# 740043) [21], indicating that pure-phase Cs_2_NaYF_6_:15Tb NPs have been successfully prepared. The crystal structure of Cs_2_NaYF_6_:Tb is shown in Figure 1c. Cs_2_NaYF_6_ is a compound with a cubic perovskite-derived structure, and its crystal structure is crystallized in the Fm′3m space group. In this structure, Cs^+^ ions combine with 12 equivalent F^−^ to form a CsF_12_ cube; Na^+^ and Y^3+^ combine with 6 equivalent F^−^ to form NaF_6_ and YF_6_ octahedra, respectively. The doped Tb^3+^ partially replaces the Y^3+^ ion. Inductively coupled plasma optical emission spectroscopy (ICP-OES) analysis indicates that the Tb^3+^ doping concentration is approximately 14.2 mol%, which is consistent with the nominal value.

The particle shown in Figure 2a is composed of many smaller particles, which facilitates clear elemental mapping. The elemental distributions of Cs, Na, Y, F, and Tb are shown in Figure 2b–f, indicating a homogeneous distribution of these elements throughout the synthesized sample. As shown in Figure 2g, energy-dispersive X-ray spectroscopy (EDS) profiling confirmed the presence of Cs, Na, Y, F, and Tb elements in the Cs_2_NaYF_6_:15Tb sample. Combined with the XRD analysis, these results verify the successful synthesis of the target product with the expected chemical composition and crystalline structure. Upon X-ray irradiation, the Cs_2_NaYF_6_:15Tb nanoparticles exhibit intense green luminescence, with characteristic emission peaks at 490, 543, 586, and 621 nm, corresponding to the Tb^3+^: ^5^D_4_ → ^7^F_J_ (J = 6, 5, 4, and 3) transitions (Figure 2h) [24]. In addition, even after continuous X-ray irradiation at 50 kV for 10 min, no afterglow signals were detected (Appendix A).

### 3.2. OA Content and [Cs^+^]/[Na^+^] Ratio Effects on NP Structure

The influences of the [Cs^+^]/[Na^+^] ratio and OA content on the structure of as-prepared NPs were studied. As shown in Figure 3a, when the [Cs^+^]/[Na^+^] ratio was 6:1 and 8:1, both the cubic YF_3_ and Cs_2_NaYF_6_ phases were formed. When the [Cs^+^]/[Na^+^] ratio increases to 10:1 and 12:1, each XRD diffraction peak of the prepared NPs was in agreement with cubic Cs_2_NaYF_6_ (JCPDS# 740043). As shown in Figure 3b, with increasing OA content in the precursors solution, impure cubic YF_3_ NPs were formed. In addition, the morphology was greatly changed with an increase in OA content from 8 to 13 mL, and then the mean particle size was evidently decreased with further increasing of the OA content to 20 mL (Appendix A). Hence, a relative lower OA content and higher [Cs^+^]/[Na^+^] ratio benefit the formation of pure Cs_2_NaYF_6_ NPs. Considering that the XEOL intensity for the case of 10:1 was higher than that of 12:1 (Appendix A), the [Cs^+^]/[Na^+^] ratio of 10:1 was used to prepare the other NPs studied in this work.

### 3.3. Tb^3+^ Doping Optimization in Cs_2_NaYF_6_ NPs

Figure 4a shows the XRD patterns of Cs_2_NaYF_6_:xTb (x = 5~25%) NPs, and the diffraction peaks of all samples were exactly matched to the JCPDS# 740043 standard card, and no impurity peaks were detected, indicating that pure-phase Cs_2_NaYF_6_ NPs were successfully synthesized. Figure 4b,c and Appendix A illustrate the luminescence performance of Cs_2_NaYF_6_:xTb (x = 5~25%) NPs under X-ray and 254 nm UV excitation, respectively. With an increase of Tb^3+^ doping concentration from 5 mol% to 20 mol% under X-ray excitation, the XEOL intensity increased significantly, which was attributed to the increase in the number of luminescence centers. However, at concentrations above 20 mol%, the luminous intensity decreases due to the cross-relaxation effect between Tb^3+^. Therefore, the optimal doping concentration under X-ray excitation is 20 mol%, where the XEOL integration intensity of Cs_2_NaYF_6_: 20Tb NPs is 7.6 higher than that of the 5% doped sample (Figure 4d). The emission peaks are located at 480 nm, 545 nm, 585 nm, and 620 nm, corresponding to the ^5^D_4_→^7^F_6_, ^5^D_4_→^7^F_5_, ^5^D_4_→^7^F_4_, and ^5^D_4_→^7^F_3_ transitions of Tb^3+^, respectively. Their full widths at half maximum (FWHM) were about 13.5, 3.7, 9.0, and 7.1 nm, respectively. Upon continuous X-ray irradiation for 1 h, the emission intensity of Cs_2_NaYF_6_:Tb showed no significant change (Appendix A). Under the excitation of 254 nm ultraviolet light, the luminous intensity of Tb^3+^ increased from 5 mol% to 15 mol% but decreased at 20 mol% due to the concentration quenching effect. The optimal doping concentration was 15 mol%, and the luminous intensity of Cs_2_NaYF_6_:15Tb was increased by 3.6 times compared with the 5% doped sample (Figure 4d). Hence, the optimal doping concentrations under X-ray and ultraviolet excitation were 20 mol% and 15 mol%, respectively. This difference stems from the difference between the two excitation mechanisms: X-rays excite Tb^3+^ by interacting with the matrix to produce secondary electrons, while UV light directly excites the 4f-5d transition of Tb^3+^. As the Tb^3+^ concentration increases, the average distance between activator ions decreases, thereby increasing the probability of non-radiative energy transfer processes such as cross-relaxation between neighboring Tb^3+^ ions [25]. This process leads to depopulation of the excited states responsible for emission, ultimately resulting in reduced luminescence intensity. This study provides an important basis for the regulation of luminescence properties of rare-earth-doped nanomaterials, especially in multi-mode excitation applications.

### 3.4. Luminescence Performance of Cs_2_NaYF_6_ NPs with Eu^3+^, Dy^3+^, Er^3+^, and Tm^3+^ Doping

We further investigated the effect of doping with other lanthanide ions (Eu^3+^, Dy^3+^, Er^3+^, and Tm^3+^) on the Cs_2_NaYF_6_ matrix. Figure 5a shows that the XRD diffraction peaks of Cs_2_NaYF_6_:10Eu, Cs_2_NaYF_6_:0.5Dy, Cs_2_NaYF_6_:2Er, and Cs_2_NaYF_6_:0.5Tm NPs are matched to the JCPDS# 740043 standard card without stray peaks, indicating that the pure-phase material has been successfully prepared. Figure 5b,c and Appendix A show the XEOL spectra of each doped sample under X-ray excitation: Cs_2_NaYF_6_:10Eu exhibits a ^5^D_0_→^7^F_1_ transition emission of Eu^3+^ at 594 nm; Cs_2_NaYF_6_:0.5Dy shows a ^4^F_9/2_→^6^H_13/2_ transition of Dy^3+^ at 582 nm; Cs_2_NaYF_6_:2Er and Cs_2_NaYF_6_:0.5Tm produce characteristic emission peaks at 403 nm (Er^3+^: ^2^H_9/2_→^4^I_15/2_) and 448 nm (Tm^3+^: ^1^G_4_→^3^H_6_), respectively [26,27,28,29,30]. Among them, the XEOL integration intensity of Cs_2_NaYF_6_:10Eu was significantly higher than that of other systems, and its intensity was 25.9 times that of Cs_2_NaYF_6_:0.5Dy (Figure 5d). These results show that the Cs_2_NaYF_6_ matrix can effectively adapt to the luminescence characteristics of different lanthanide ions, and Eu^3+^ exhibits the best X-ray excitation luminescence performance due to its unique electronic structure and high-efficiency energy level transition.

### 3.5. The XEOL Mechanism of Cs_2_NaYF_6_:Ln NPs

The XEOL mechanism for Cs_2_NaYF_6_:Ln NPs is illustrated in Figure 6. Upon X-ray irradiation, the photoelectric effect and the Compton scattering effect occur, which produce hot electrons and deep holes. Then, a large number of secondary electrons are generated due to electron–electron scattering and the Auger process, resulting in low-kinetic charge carriers. In this process, many low-energy electrons and holes gradually accumulate in the conduction and valence bands. After the electron populations on the 5d and higher 4f energy levels of lanthanide activators, the typical 4f-4f transitions are recorded [31].

### 3.6. Comparison of XEOL Intensities Between Cs_2_NaYF_6_:15Tb and NaYF_4_:15Tb NPs

We compared the XEOL intensities of cubic phase NaYF_4_:15Tb and Cs_2_NaYF_6_:15Tb NPs. As shown in Figure 7a, the XRD profile shows that the diffraction peaks of the prepared NaYF_4_:15Tb NPs are a complete match to the standard card for cubic phase NaYF_4_ (JCPDS# 77-2042) [17], and no impurity peaks are observed, confirming its pure-phase structure. As shown in Figure 7b, the XEOL intensity of Cs_2_NaYF_6_:15 Tb NPs is significantly higher than that of NaYF_4_:15Tb NPs. This phenomenon is mainly attributed to the introduction of Cs^+^ with a plateau molecular number (Z) into the NaYF_4_ matrix, which significantly enhances the absorption capacity of the material for X-ray. Specifically, the high-Z characteristics of the Cs^+^ allow it to absorb X-ray photons more efficiently and transfer their energy to the Tb^3+^ luminescence center, thereby significantly improving luminescence efficiency. To further avoid the influence of surface quenchers on XEOL intensity, we treated the NaYF_4_:15Tb NPs at 300 °C. As shown in Appendix A, after being treated at a high temperature, the XEOL intensity is slightly enhanced, while it remained lower than that of Cs_2_NaYF_6_:15Tb. These results show that the Cs_2_NaYF_6_ matrix is superior to the NaYF_4_ matrix in terms of X-ray excitation luminescence performance, which provides an important basis for the development of high-efficiency X-ray scintillator materials.

### 3.7. High-Resolution X-Ray Imaging Capability of Cs_2_NaYF_6_:15Tb NPs

As shown in Figure 8a, the XEOL intensity of Cs_2_NaYF_6_:15Tb NPs exhibits a good linear relationship with the X-ray dose rate, indicating that they can still provide clear imaging signals at low doses, which is an important feature of high-performance scintillator materials. To evaluate their imaging capabilities, Cs_2_NaYF_6_:15Tb NPs were prepared into homogeneous films and imaged on standard X-ray resolution test pattern plates [32]. As shown in Figure 8b,c and Appendix A, the resolution of the thin film was as high as 20.0 LP/mm; this spatial resolution exceeds that of most commercial halide perovskites [33] (typically less than 10 LP/mm) and CsI (Tl) [34] (≈10 LP/mm), which fully demonstrates its excellent X-ray imaging performance. In addition, we used this film to X-ray image a melon seed, and the results are shown in Figure 9d. The imaging image clearly shows the shape and contour of the melon seed kernel and the details of its internal structure, indicating that the Cs_2_NaYF_6_:15Tb nanofilm can accurately capture X-ray imaging information of the microstructure. These experimental results confirm the potential of Cs_2_NaYF_6_:15Tb NPs in high-resolution X-ray imaging.

### 3.8. Dual-Mode X-Ray/UV–Excitable Cs_2_NaYF_6_:Tb NPs for High-Sensitivity Spermine Detection

Cs_2_NaYF_6_:20Tb NPs were dispersed in deionized water, and spermine standard solutions (0–60 μM) were incrementally added for X-ray and ultraviolet (UV) spectroscopic measurements. As shown in Figure 9a,c, the luminescence intensities of Cs_2_NaYF_6_:Tb NPs gradually decrease with increasing spermine concentrations under X-ray and 254 nm UV excitation. This quenching effect is attributed to spermine adsorption on the nanoparticle surface, which introduces additional non-radiative relaxation pathways for Tb^3+^, thereby reducing luminescence intensity [35,36,37,38]. To assess the probe’s performance, the limit of detection (LOD) was analyzed. The spermine concentration and fluorescence intensity data in the range of 0–60 μM were fitted based on the following equation:LOD = 3σ/K
where σ is the standard deviation of the blank value [37,38], and K is the slope of the calibration curve. The LOD values were determined to be 0.44 μM under X-ray excitation and 0.36 μM under UV excitation (Figure 9b,d). These results indicate that the probe achieves submicromolar detection sensitivity, making it suitable for biomedical testing applications.

## 4. Discussion

Cs_2_NaYF_6_:Tb NPs were successfully prepared by co-precipitation, and the controllable preparation of pure-phase materials was achieved by adjusting the molar ratio of Cs^+^/Na^+^ (10:1) and the amount of oleic acid (8 mL). By optimizing the doping concentration of rare earth ions, it was found that the optimal doping concentrations of Tb^3+^ were 20 mol% and 15 mol% under X-ray excitation and ultraviolet excitation, respectively. The Cs_2_NaYF_6_:15Tb film exhibited excellent X-ray response characteristics with a spatial resolution of 20 LP/mm. Beyond X-ray imaging applications, Cs_2_NaYF_6_:Tb NPs were also explored for spermine detection, demonstrating high sensitivity with a detection limit of 0.44 μM (under X-ray excitation) and 0.36 μM (ultraviolet excitation) within a concentration range of 0–60 μM.

## Figures and Tables

**Figure 1 nanomaterials-15-00728-f001:**
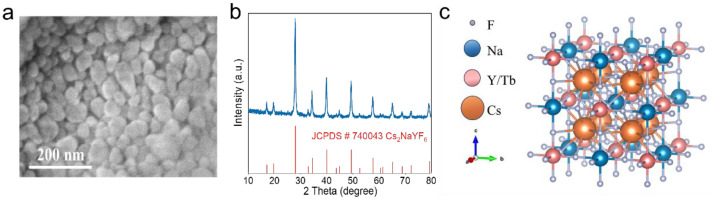
SEM image (**a**), XRD pattern (**b**), and schematic illustration of the crystal structure (**c**) of Cs_2_NaYF_6_:15Tb NPs.

**Figure 2 nanomaterials-15-00728-f002:**
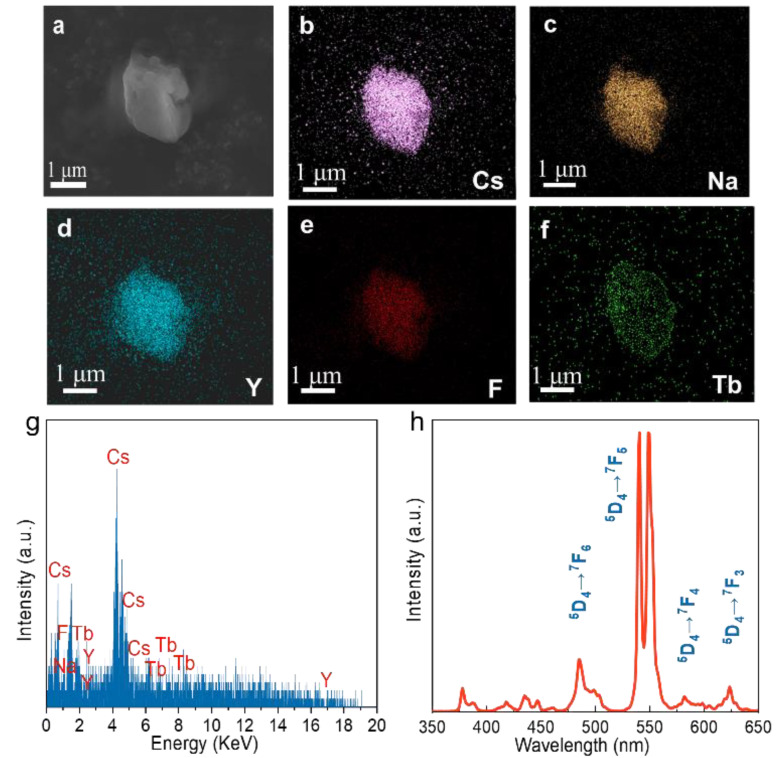
(**a**–**f**) SEM image and element mapping results, (**g**) EDX spectrum, and (**h**) XEOL spectrum of the Cs_2_NaYF_6_:15Tb NPs.

**Figure 3 nanomaterials-15-00728-f003:**
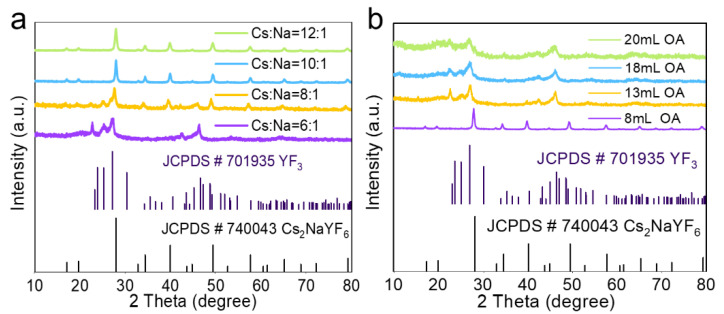
(**a**) XRD patterns of the Cs_2_NaYF_6_:15Tb prepared with different [Cs^+^]/[Na^+^] ratios. (**b**) XRD patterns of the Cs_2_NaYF_6_:15Tb prepared with different oleic acid contents. JCPDS# 740043 and 701935 represent the standard data of cubic Cs_2_NaYF_6_ and YF_3_, respectively.

**Figure 4 nanomaterials-15-00728-f004:**
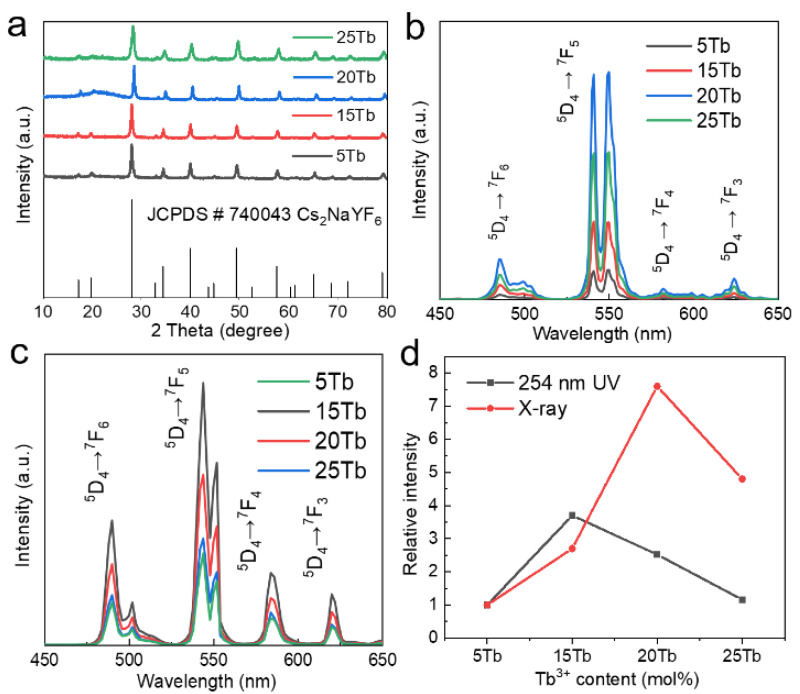
(**a**) XRD patterns and (**b**) XEOL spectra of Cs_2_NaYF_6_:xTb (x = 5~25%) NPs. (**c**) Photoluminescence spectra of Cs_2_NaYF_6_:xTb (x = 5~25%) NPs acquired under UV excitation. (**d**) Comparison of the integrated intensities of XEOL spectra and ultraviolet excitation luminescence spectra of Cs_2_NaYF_6_:xTb (x = 5~25%) NPs.

**Figure 5 nanomaterials-15-00728-f005:**
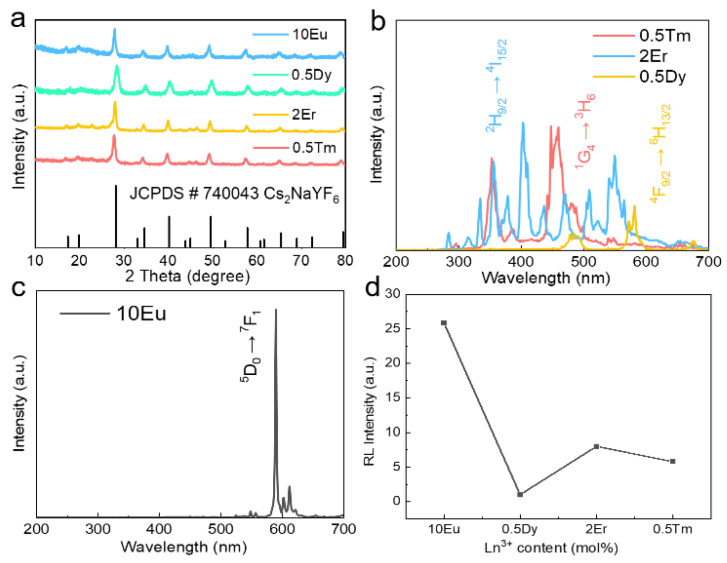
(**a**) XRD patterns of Cs_2_NaYF_6_:10Eu, Cs_2_NaYF_6_:0.5Dy, Cs_2_NaYF_6_:2Er, and Cs_2_NaYF_6_:0.5Tm NPs; (**b**) XEOL spectra of Cs_2_NaYF_6_:0.5Tm, Cs_2_NaYF_6_:2Er, and Cs_2_NaYF_6_:0.5Dy NPs. (**c**) XEOL spectra of Cs_2_NaYF_6_:10Eu NPs. (**d**) Integral XEOL intensity comparison of Cs_2_NaYF_6_:10Eu, Cs_2_NaYF_6_:0.5Dy, Cs_2_NaYF_6_:2Er, and Cs_2_NaYF_6_:0.5Tm NPs.

**Figure 6 nanomaterials-15-00728-f006:**
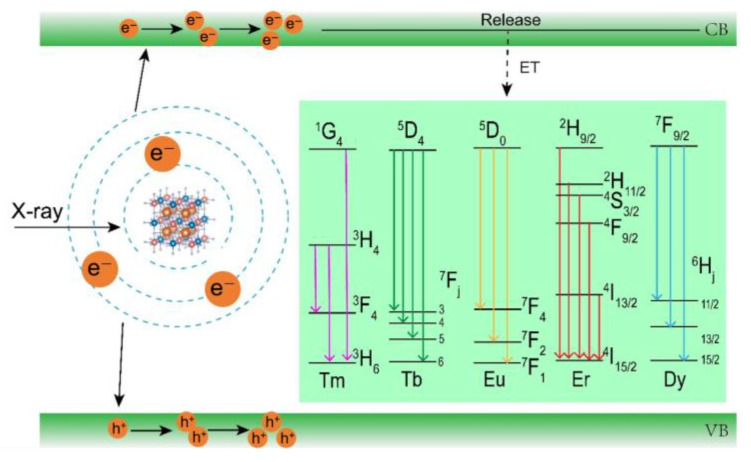
Mechanistic investigations of the XEOL of Cs_2_NaYF_6_:Ln NPs.

**Figure 7 nanomaterials-15-00728-f007:**
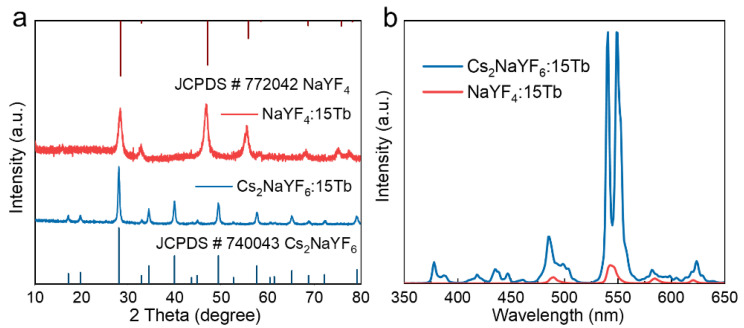
(**a**) XRD patterns of NaYF_4_:15Tb and Cs_2_NaYF_6_:15Tb NPs. (**b**) Comparative XEOL spectra of NaYF_4_:15Tb and Cs_2_NaYF_6_:15Tb NPs.

**Figure 8 nanomaterials-15-00728-f008:**
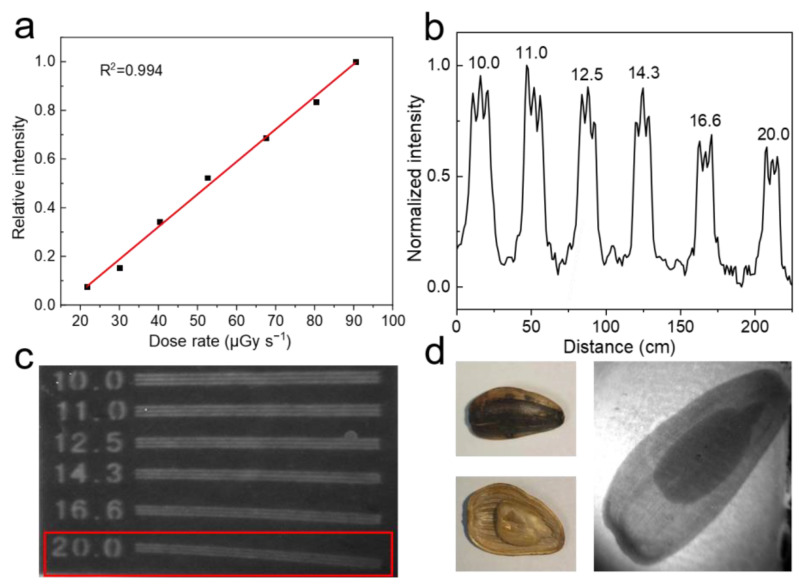
(**a**) Relationship between XEOL intensity of Cs_2_NaYF_6_:15Tb NPs and X-ray dose rate; (**b**) gray value function of real-time X-ray imaging with a resolution range of 10.0–20.0 LP/mm; (**c**) XEOL imaging of a standard X-ray resolution test pattern; (**d**) the internal and external structure of a melon seed and its XEOL image. X-ray dose: 0.5 Gy.

**Figure 9 nanomaterials-15-00728-f009:**
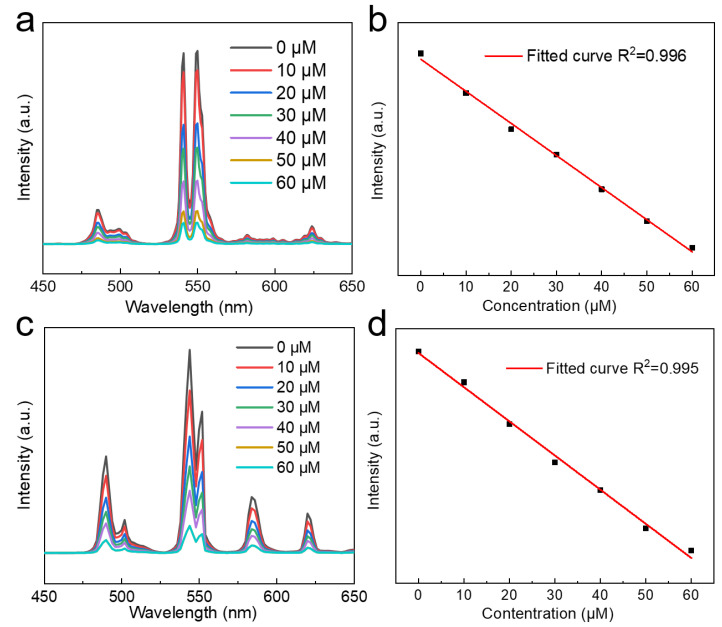
(**a**) XEOL spectra of Cs_2_NaYF_6_:20Tb NPs at different spermine concentrations under X-ray excitation. (**b**) Linear relationship between the integral luminous intensity of Cs_2_NaYF_6_:20Tb NPs and spermine concentration under X-ray excitation. (**c**) Photoluminescence emission spectra of Cs_2_NaYF_6_:15Tb NPs at different spermine concentrations under 365 nm ultraviolet excitation. (**d**) Linear relationship between the luminous intensity of Cs_2_NaYF_6_:15Tb NPs and spermine concentration under 365 nm UV excitation.

## Data Availability

The data that support the findings of this study are available from the corresponding author upon reasonable request.

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
