# Peer review of "Controlled Synthesis of Cs2NaYF6: Tb Nanoparticles for High-Resolution X-Ray Imaging and Molecular Detection"

_nanomaterials, 2025, doi:10.3390/nano15100728_

Round 1
Reviewer 1 Report (Previous Reviewer 2)
Comments and Suggestions for Authors
I have no further comments for this manuscript.
Author Response
I really appreciate your suggestion to publish our work.
Reviewer 2 Report (Previous Reviewer 3)
Comments and Suggestions for Authors
The authors have addressed my previous comments and made meaningful improvements to the manuscript. I appreciate that they conducted additional experiments to further support their conclusions. I have only one minor suggestion: the authors used ICP-OES to determine the Tb concentration in their sample. Please include detailed information on the instrumentation, what standard was used, and specific sample preparation procedures in the Experimental section.
Author Response
According to the reviewer’s comment, the following contents have been added in the Experimental section.Inductively coupled plasma optical emission spectrometry (ICP-OES) is per-formed using an Agilent 720ES instrument, equipped with a high-frequency generator operating at 27.12 MHz to sustain a stable plasma. Calibration is carried out using a series of terbium standard solutions, prepared by appropriate dilution in 2% v/v nitric acid to generate a concentration-dependent calibration curve. All standards are pre-pared in volumetric flasks and mixed thoroughly to ensure homogeneity. Sample Preparation Procedures: 0.5 g sample was added into 10 mL nitric acid. The resulting solution is transferred to a 20 mL volumetric flask. The digestion vessel is rinsed mul-tiple times with deionized water, and the rinse solutions are added to the flask.
Reviewer 3 Report (Previous Reviewer 4)
Comments and Suggestions for Authors
The article looks good after the corrections have been made. In my opinion the article is ready for publication.
Author Response
I really appreciate your suggestion to publish our work.
This manuscript is a resubmission of an earlier submission. The following is a list of the peer review reports and author responses from that submission.
Round 1
Reviewer 1 Report
Comments and Suggestions for Authors
The manuscript presents the synthesis and characterization of Cs₂NaYF₆:Tb³⁺ nanoparticles with promising performance in high-resolution X-ray imaging and molecular sensing applications. The topic is interesting, timely, and relevant to the field of nanomaterials for optoelectronic applications. The experimental approach is generally sound, and the results are significant. However, I have several comments and questions that the authors should address before publication:
- The authors mention that incorporating Cs⁺ (a high-Z ion) improves XEOL performance. How does this specific Cs₂NaYF₆:Tb system compare to other recent high-Z host materials (e.g., Ba-based or Pb-based fluoride systems)? Could the authors briefly position their results against other high-density hosts in terms of light yield, stability, or toxicity?
- Consider emphasizing more clearly what the major novelty is compared to prior works on Cs₂NaYF₆ materials (e.g., the cited work by Yang et al., Ref [20]).
- In the Materials and Methods section, could the authors explain why the specific ratios of solvents (OA, OLA, ODE) were chosen? Were different solvent systems or surfactant ratios tested?
- How reproducible is the synthesis? Were multiple batches tested for uniformity in morphology and luminescence?
- In Figure 1, the SEM shows irregular morphology. Could the authors provide a size distribution histogram from multiple nanoparticles to better assess uniformity?
- Have the authors performed any quantitative elemental analysis (such as ICP-OES or XPS) to confirm the doping level of Tb³⁺?
- XEOL spectra are shown, but did the authors measure absolute light yield (e.g., in photons/MeV) compared to a standard scintillator?
- What is the photostability of the Cs₂NaYF₆:Tb nanoparticles under prolonged X-ray exposure? Any signs of degradation?
- In Figure 2h, what is the full width at half maximum (FWHM) of the emission peaks?
- In the doping optimization (Section 3.3), the optimal doping for X-ray excitation is 20%, but for UV excitation it's 15%. What would be the behavior under simultaneous dual-mode excitation? Could this dual-mode response be useful?
- Could the authors comment on the physical mechanism of concentration quenching observed at higher Tb³⁺ content? Is it energy transfer, cross-relaxation, or clustering?
- The decrease in XEOL intensity upon Li⁺ doping is attributed to morphology or lattice symmetry changes. Have the authors performed any additional structural characterization (e.g., Raman spectroscopy, XPS) to verify this hypothesis?
- The reported resolution of 20 lp/mm is excellent. Could the authors compare the light output per dose (sensitivity) quantitatively with CsI:Tl and halide perovskite scintillators?
- What was the X-ray dose used during imaging experiments? Can the authors specify whether the material exhibits afterglow, which could affect imaging quality?
Reviewer 2 Report
Comments and Suggestions for Authors
The manuscript by Jian et al. reports the development of Cs2NaYF6:Tb NPs and their use in X-ray imaging and spermine detection applications. This system and similar systems have been widely reported. While, I do not see any obvious new or significant features from this manuscript in terms of material synthesis, optical properties, and application prospects.
- This manuscript is not well written and carefully checked, filled with numerous grammatical errors, incomplete sentences, and missing information.
- The captions of Figure.3, Figure.4, Figure S1 are wrongly labelled. The images and text don't match at all.
- According to the authors’ theory that using high-Z Cs+ ions to increase the XEOL intensity. There is no reason to use the Cs2NaYF6 matrix. Cs3YF6, Cs2NaLuF6, Cs3LuF6 will be better. And their synthesis will be very similar. For the synthesis, why did the author use different types of Li/Na/Cs precursors, for example NaAc, LiAc, CsF? Why does the authors study the impact of OA, not other precursor like OLA?
- Fig.1a shows that the prepared NPs are amorphous, instead of the cube-like morphology that the cubic phase should have, as seen in perovskites. This indicates that the NPs growth process is not well controlled. The imperfection of the structure will inevitably affect the performance of the material.
- Fig.1a shows that the size of the NPs is about 100 nm, while Figure 2 shows that close to 2 μm.
- Figure 2e does not indicate the corresponding element.
- We suggest the authors to check the EDX data carefully and use the pictures with caution, as I noticed that the NP shapes of different elements are not from the same particle.
- Why do the authors only analyze impurities by comparing to YF3? Other fluorides NaYF4, TbF3, NaTbF4… should all be considered.
- It is strange why the authors suddenly want to study the effect of Li+ doping. Since the scientific problem is not explained at all, I tend to suspect that the author's logic is unclear and does not know what they are doing, or they are just piling up experimental data.
- For the sentence “After the addition of Li+, the XEOL intensity especially in the UV region was decreased (Fig. 7b), which probably attributed to the changed morphology or lattice symmetry.” How does the changed morphology or lattice symmetry affect optical performance?
- For the comparison of XEOL Intensities between Cs2NaYF6:15Tb and NaYF4:15Tb NPs, the particle size will also influence the XEOL intensities, so the authors should provide the sizes and justify this comparison.
- Why do the authors perform spermine detection? “This quenching effect is attributed to spermine adsorption on the nanoparticle surface, which introduces additional non-radiative relaxation pathways for Tb3+, thereby reducing the luminescence intensity.“ The author does not provide any evidence, such as IR spectra.
The authors should revise their writing and scientific presentation.
Reviewer 3 Report
Comments and Suggestions for Authors
Zhao et al presented a study comparing the luminescence properties of Cs2NaYF6-based nanoparticles. The study addresses a potentially valuable topic, and with careful revision and improved discussion, it could make a meaningful contribution to the field. Specific issues should be addressed are outlined below:
- introduction
- The language needs to be polished, particularly the introduction part. For example, in the second paragraph, phrases like “exhibited facile synthesis procedure”, “”perform much high stability”, “up to date”, etc were not grammatically correct. The sentence “Lanthanide activators possess abundant excited energy levels, enabling broad emissions …” is misleading. Usually activators with many energy levels lead to narrow emission lines, instead of broad emission bands.
- Please double-check the definitions of symbols for the X-ray attenuation coefficient expression. “A is atomic mass” is incorrect; it should be atomic mass unit. If Z represents an atomic number, which atom should be used to calculate the coefficient for a compound? With Z undefined, it is difficult to interpret the expression that later occurred in the text: NaLuF4:15Tb (Z=71), why Z=71?
- There are multiple places where the sample name included an expression “15Tb” throughout the entire manuscript. It is unclear what “15” means. Perhaps it means doping concentration, but it wasn’t defined until a much later section where dopant concentration was discussed.
- Experimental
- In the first paragraph, it was mentioned “x mmol” of Y(Ac)3, and “(1-x) mmol” of Ln(Ac)3. The value of x was never mentioned. Later in section 3.3, “x” was re-defined as the concentration of Tb, which is inconsistent with the previous expression.
- The first two paragraphs of the experimental can be combined into one, because almost all the steps are identical, only some precursors were different.
- Please include more details about the X-ray source used, e.g. what is the energy? How was the X-ray imaging obtained?
- Results and discussion
- Please use the correct format when writing the JCPDS #.
- The statement “This structure is characterized by a high degree of symmetry and order, which makes Cs2NaYF6 a potential application in the fields of optics and electronics.” Is unsupported. Why is a crystal with high symmetry and order potentially good for optics and electronics?
- There is a discrepancy about the particle size. In the SEM image (Figure 1) the particles are ~50 nm, but in Figure 2, which says “an individual particle” in its caption, clearly has a particle larger than 1 μm.
- Figure 2e is missing the element symbol. In general, the EDX spectrum is not convincing that Y and Tb are present in the sample. What were the atomic percentages of these elements?
- Figure 3a is the Cs/Na ratio study, but the text says it was the OA content study. Please revise.
- Figure 4 caption. What does “normalized integral XEOL intensities” mean in (b)? 4(c) is photoluminescence spectra acquired under UV excitation, not UV spectra.
- Concern about Section 3.4. This section compared the luminescence of different dopants, and intensity was one of the main parameters being compared. However, such a comparison is not valid, because the dopant concentrations vary significantly. The brightest sample, claimed by the authors, was Cs2NaYF6:10Eu, while other dopants were all less than 2%. It is unclear whether the high intensity is simply due to high dopant concentration. Without further supporting evidence, it can not be related to the underlying energy transfer mechanism.
- Section 3.6, the last sentence is vague and unsupported. The authors attributed lower luminescence upon Li doping to “probably changed morphology or lattice symmetry”. Please elaborate.
- Section 3.9. The quenching effect was attributed to molecular adsorption. This is superficial and unsupported. Is this adsorption-related quenching only specific to spermine? Or will any molecule quench the luminescence? What exactly was the chemical interaction?
Reviewer 4 Report
Comments and Suggestions for Authors
I revised manuscript no. nanomaterials-3605381 titled: “Controlled synthesis of Cs2NaYF6: Tb nanoparticles for high 2 resolution X-ray imaging and molecular detection” applied as an “Article” article type.
I appreciate the bimodal application of novel material as a scintillator and for optical spermine detection. The idea for the research is interesting. The research is done well, and the article is written correctly. I recommend to accept this article after major corrections and solving a few queries:
- Why cubic Cs2NaYF6? The most common fluoride material is hexagonal NaYF4, why Cs2NaYF6 is superior for such properties? Please add a comparison of different materials and their XEOL properties in the table.
- Please correct the units in paper.
- “Compared to halide perovskites, lanthanide-doped fluoride NPs with covalent crystal structures perform much higher stability [10].” - fluorides are not covalent but ionic crystals.
- Y(Ac)3, ODE, OLA, OA and others should be explained in the text.
- The used method for synthesis is well-known as the thermal-decomposition method. Please add such information.
- Mentioned “PMMA toluene solution” is standard for the preparation of film for X-ray imaging purposes?
- Why do authors investigate Y3+ based matrix, while in the introduction authors highlighted that matrix with Lu3+ is better for such applications?
- What element is mapped in Fig. 2e?
- A proper baseline is important for emission spectra. Please start the intensity scale with Y = 0. Please correct Fig. 2h, 4c, 5b, 7b, 8b, 10a,c.
- 4 - More specific % between 15 and 20 may allow us to determine the optical content of Tb3+ for both applications. Why such investigations were not done?
- 5b - spectra are not presented properly, some bands are hard to distinguish. Not all transitions are described.
- The authors used different Ln3+ dopants (Eu, Dy, Tm, Er). Content of Ln was optimized, based on some research, or just random?
- Authors used also Li+ dopant. Why? The doping of Li+ should be explained. What was the motivation? How does Li+ influence crystal structure and photoluminescence/XEOL properties?
- NaYF4 and Cs2NaYF6 doped with Tb3+ were compared. Did the author perform Tb-content optimization for NaYF4? Optimal concentration can be different for different materials.
- 8a - Size and crystallinity of the NaYF4 and Cs2NaYF6 NPs are clearly different.
- “Cs2NaYF6:15Tb NPs were prepared into homogeneous films and imaged on standard X-ray resolution test pattern plates.” The test should be explained in the methodology. Citation should be added.
- Please add a high-resolution photo of Fig. 9c to SI.
- Photoluminescence spectra should not be described only as “emission spectra” but as photoluminescence emission spectra (PL spectra), because XEOL is also a kind of emission.
- Only one citation style should be maintained.
- Please transfer methods to the main article from SI.